# Abu Dhabi Neural Mapping (ADNM) during Minimally Invasive Thyroidectomy Enables the Early Identification of Non-Recurrent Laryngeal Nerve and Prevents Voice Dysfunction

**DOI:** 10.3390/jcm11195677

**Published:** 2022-09-26

**Authors:** Iyad Hassan, Lina Hassan, Ibrahim Gamal, Mohamad Ibrahim, Abdel Rahman Omer

**Affiliations:** 1Department of Surgery, Burjeel Hospital, Abu Dhabi 7400, United Arab Emirates; 2Department of Anesthesia and Pain Management, Burjeel Hospital, Abu Dhabi 7400, United Arab Emirates

**Keywords:** voice, neuromonitoring, vocal cord paresis, non-recurrent laryngeal nerve, thyroidectomy

## Abstract

The aim of this study was to evaluate the diagnostic accuracy of a neuromonitoring protocol—the Abu Dhabi Neural Mapping protocol (ADNM)—using a new device, Nim-Vital™, during minimally invasive thyroidectomy in the early identification of non-recurrent laryngeal nerve (n-RLN) problems and the preservation of its function. Method: Patients with thyroid disorders that required thyroid resection, who were admitted to the Department of Surgery at Burjeel Hospital, Abu Dhabi, between January and July 2022, were included in the study. The data were extracted from a prospective database and were analyzed retrospectively. All nerves at risk were identified and exposed at seven precisely defined anatomical points, with strict adherence to the intraoperative technical steps of neuromonitoring. These were sequentially applied to the vagal nerve (VN), the superior laryngeal nerve (SLN), and the recurrent laryngeal nerve (RLN). In the next step after the creation of the skin-platysma flap, the strap muscle’s lateral border was moved from the medial limb of the sternocleidomastoid without using any electrical device and without any manipulation of the thyroid gland. The VN was exposed in the carotid sheath and then stimulated using a monopolar probe at a precisely defined point above the clavicle, using anatomical landmarks. Results: In total, 136 women with a mean age of 40 years (range 18–74) and 36 men with a mean age of 42 (range 21–66), demonstrating 270 nerves at risk, were included in the analysis. Indications for surgery were malignancy in 70 cases, toxic goiter/Graves in 23 cases, retrosternal goiter in 21, and symptomatic multinodular goiter in 64 cases. Of these, 100 patients received a total thyroidectomy, 46 received a right lobectomy, and 24 received a left lobectomy only. For a total thyroidectomy, the median skin-to-skin surgery duration was 52 min (range 24–104 min) and the median hospital stay was 2 days (range 1–4 days). In 4 cases (4/146; 2.74%) the pre-dissection stimulation of the vagal nerve (VN1) at the ADNM’s precisely defined point did not create any signal or proper EMG-curve that indicated the existence of the non-RLN. Proximal dissection of the right VN at a precisely defined point by the ADNM’s level of incisura of the larynx created a positive signal. The separation point of the right non-RLN from the VN was discovered in all four patients. The postoperative video-laryngoscopy confirmed bilateral mobile vocal cords in all cases. Conclusions: Following the ADNM protocol during thyroid surgery minimizes the risk of a non-laryngeal nerve injury and prevents voice dysfunction.

## 1. Introduction

The incidence of differentiated thyroid cancer is rapidly increasing around the globe and, with a crude prevalence of 11% in the United Arab Emirates (UAE), became the second most common type of cancer in women in the UAE. As a consequence of this so-called “thyroid cancer epidemic,” the thyroidectomy rate is increasing, which, in turn, is a major cause of unilateral vocal fold paralysis (UVFP). Currently, the percentage of postoperative vocal fold nerve injuries is estimated to be between 1% and 30% for transient lesions and between 0.5% and 5% for permanent ones. UVFP is responsible for disabling dysphonia, low quality of life, and a range of medico-legal implications [1].

A non-recurrent laryngeal nerve (non-RLN) is a rare and more complex anatomical variant of the recurrent laryngeal nerve (RLN) [2,3]. It has an aberrant pathway that directly enters the larynx without first descending into the thorax [4]. It is mainly found on the right side of the body, with an incidence rate of 0.5–1%, whereas the left-sided non-RLN is extremely rare [5]. The presence of a non-RLN makes a person prone to injury during thyroidectomy and leads to severe postoperative complications if not treated carefully. It has been reported previously that the occurrence of postoperative nerve injury increases by six times if a non-RLN remains undetected [6].

Intraoperative neuromonitoring has been gaining popularity as a supplement to visual identification, due to its usefulness in detecting RLN during thyroid and parathyroid surgery and preventing injury [7]. However, when the surgeon cannot detect an RLN, he must consider that a non-RLN is present [8]. Two types of non-RLN can be identified—type 1 and type 2 [9,10]. A type-1 non-RLN travels along the superior thyroid pedicle, while type 2 follows the path of the inferior thyroid artery [8]. During thyroidectomy, both types must be identified; the best method to detect them is via intraoperative neuromonitoring [8]. A neck CT scan or MRI can identify the vessel’s position. However, this raises the cost of the operation significantly and also incorporates a high radiation dose [8]. Intraoperative neuromonitoring can be a suitable alternative in this regard. It is associated with a higher identification rate of non-RLN than any other method, at almost 6% [8].

The proper identification of non-RLN can be followed by a successful thyroidectomy where nerve functionality is restored after the surgery. Using intraoperative neuromonitoring, surgeons can retrace their surgical path so that no injury befalls a nerve [11]. Therefore, this study is the first to describe the clinical use of the NIM Vital^TM^ neuromonitoring system (Medtronic Xomed, Inc., Jacksonville, FL, USA). Furthermore, the ADNM technique was performed at precisely defined anatomical positions in a large patient cohort without the use of a neuromuscular blockade for endotracheal intubation, which enabled the easy identification of right-sided non-RLN during thyroidectomy and the prevention of voice dysfunction after surgery. The narrow operative field of a minimally invasive open thyroidectomy and the absence of neuromuscular blockade restrict the degree to which surgical instruments can be moved. These kinds of facts drive surgeons to create more novel surgical procedures in order to limit the risk of vocal nerve injury that is associated with serious voice disorders.

The small operational field in a minimally invasive open thyroidectomy, as performed, and not using any neuromuscular blockade limiting instrument-maneuvering degree offers surgeons the opportunity for more innovative surgical techniques that can avoid any neural injury and prevent major voice disorders.

## 2. Method

### 2.1. General Anaesthesia without Neuromuscular Blockade for ADNM-Associated Minimally Invasive Open Thyroidectomy

After positioning the patient on the operation table, the placement of a pulse oximeter, ECG electrode, and blood pressure cuff, and the recording of baseline vital signs were conducted. Then, a cannula was positioned. After that, IV fluid was injected, then an injection of 2 mg midazolam was administered. An oxygenation mask was kept just above the face of the spontaneously breathing patient for pre-oxygenation, with continuous control of the saturation until it reached 100%. Then, an injection of 20 mg lidocaine, followed by propofol injection as a bolus (dose calculation 2.5 mg/kg), as well as remifentanil (0.1 mg/kg bolus dose) was administered. At this point, ventilation was provided using the anesthesia face mask, Comfort Star^®^,Dräger medical AG, Lübeck, Germany which is usually present for 1–3 min. This was followed by placement of the electromyogram endotracheal tube (EMG NIM^®)^ with visual monitoring using video laryngoscopy (C-MAC^®^, Storz), to ensure the correct positioning of the EMG electrodes between both vocal cords (Figure 1). This is crucial for intraoperative neuromonitoring. In case of difficult intubation, another propofol bolus can be given and the anesthesia can be continued with sevoflurane and remifentanil infusion.

### 2.2. Surgical Technique including ADNM for Minimally Invasive Open Thyroidectomy

A Kocher incision of 2.5–4 cm along the skin crease, positioned 2 cm above the sternal notch, was performed (Figure 2a). Before that, the skin and platysma flaps, both cranially and caudally, were prepared to separate the strap muscles from the sternocleidomastoid above the clavicle, using scissors.

Energy or electric devices should not be used at this stage. Medial retraction of the strap muscle and the thyroid lobe beneath it allows easy access to the carotid sheath and enables visualization of the middle thyroid vein and inferior thyroid artery (ITA) since all types of non-RLN are branching from the vagal nerve above the crossing point of ITA with the vagal nerve. Next, the vagal nerve (VN) was visualized in the carotid sheath and then tested via neuromonitoring below the inferior thyroid artery, just above the clavicle, to exclude the possibility of non-RLN (ADNM-VN1) (Figure 2b). The stimulation current that was used was 1 mA; the EMG response should be above 200 µV to be considered sufficient. In the case of a negative signal, the existence of non-RLN is suspected; however, to exclude any technical fault or anesthesia-related loss of signal, the same procedure was performed on the left-hand side in case of a positive signal, then back to the right side to identify the non-RLN. Next, the moving of strap muscles from the thyroid using bipolar cautery was performed. Lateral retraction of the strap muscles and the division of the Kocher lateral veins using ligaSure™ was also conducted. Careful movement of the thyroid lobe from the paravertebral fascia and then the mobilization of upper pole vessels that are very close to the thyroid gland were conducted. Testing of the superior laryngeal nerve (SLN) using NIM showed a typical curve for SLN, with a lower amplitude compared to RLN and VN (ADNM-SLN1). For the visualization and protection of SLN, the thyroid gland is grasped with a Kocher clamp very close to the upper pole vessels lifted from the prevertebral facia distally and laterally. Here we can see the SLN creating a curve from the vessels going into the larynx (Figure 3). At this point, the upper pole vessels were separated in 2–3 small lengths and were then ligated via ligaSure™ after prior testing with NIM, to exclude any neural structures from the planned ligation. Immediate cooling with 5 mL cold saline was given to prevent the spread of heat damage to the SLN and/or non-RLN. Now, the right lobe can be luxated into the small wound, which allows space to test the entire vagal nerve from the level of the incisura of the larynx to the clavicle bone. Using non-RLN, a positive signal will now be detectable at the level of the larynx incisura. Now, medial retraction of the thyroid and larynx, and lateral retraction in the case of the carotid sheath, together with the use of NIM, will help in identifying and dissecting the non-RLN, from its origin on VN until its entrance into the cricothyroid membrane. At this stage, the upper parathyroid is usually identified and preserved (Figure 4a,b). By normal anatomical RLN after mobilization of the entire lobe in the wound and by identifying the laryngeal recurrent nerve below the inferior thyroid artery, a signal was obtained. The NIM signal below ITA is documented as (ADNM-RLN-1). Identification of the lower pole vessel, which was divided using LigaSure™ after identification, and vascular preservation of the lower parathyroid was performed.

The lower part of the thyroid from the trachea was released stepwise using scissors. The mobilization of the lobe was continued with preservation of the RLN above the inferior thyroid artery, along with the documentation of RLN functionality (as ADNM-RLN2). At this stage, only vascular clips were used to avoid heat damage, along with small steps in dissection since the RLN very often branches into 2 or 3 small branches. The most anterior branch is always the one supplying the vocal cords. The dissection was performed using a fine clamp above the nerve. The bleeding was compressed with gauze and the gland was retracted by the assistant surgeon, to avoid traction-related neuropraxia (Figure 5). Testing of the RLN at the level of the Berry ligament, documented as (ADNM-RLN3), was conducted. After completion of the resection, the specimen was sent for histopathological examination.

Before closing up the situs, the anesthesia team should increase the systolic blood pressure to 150 mmHg, to detect the relevant arterial bleeding. Any bleeding at the berry ligament is stopped with vascular clips or with the careful use of a 4.0 PDS suture and endocrine loops (via 2.5-fold magnification, which is used throughout the entire procedure) to avoid RLN entrapment. After the removal of all retractors, a Valsalva maneuver was performed by an anesthesiologist until the pressure was 40 mbar, to trace any venous bleeding, particularly in the muscle that might be under the retractors. At this point, no further dissection is needed; the final neuro-monitoring, conducted for recurrent laryngeal nerve below the crossing point with ITA, was documented as ADNM-RLN4, and that for the vagal nerve at the level of the incisura of the larynx was documented as ADNM-V2. Hemostatic agents, such as fibrillar agents or others, can be used if needed. The strap muscles were closed with 3.0 Vicryl in one stitch only. Correction of dorsal neck hyperextension was performed and the platysma was closed with 3 interrupted sutures, using 4.0 rapid Vicryl. Subcuticular skin stitching and dressings were also employed. When moving the patient to PACU, the BP should not rise above 140 mmHg. The patient remains there for 2 h and can drink water. A soft diet was allowed after 6 h was allowed; parathormone and calcium from serum were provided the next day. If no clinical signs of hypocalcemia were observed, the patient was discharged. A follow-up visit was maintained in the endocrine surgery outpatient clinic after 7–10 days for wound control. Optimization of thyroid hormone—and, if needed, calcium—supplementation, as well as an ENT-video laryngoscopy for the assessment of vocal cord function, was recommended for patients.

## 3. Results and Discussion

Overall, 170 patients were enrolled in this study, with a total of 270 nerves at risk. In total, 136 women with a mean age of 40 years (range 18–74) and 36 men with a mean age of 42 (range 21–66), with 270 nerves at risk, were included in the analysis. Indications for surgery were malignancy in 70 patients, toxic goiter/Graves in 23 patients, retrosternal goiter in 21 patients, and symptomatic multinodular goiter in 64 cases. Of these, 100 patients received a total thyroidectomy, 46 patients had a right lobectomy, and 24 had a left lobectomy only. For the total thyroidectomy, the median skin-to-skin surgery duration was 52 min (range 24–104 min) and the median hospital stay was 2 days (range 1–4 days). In four cases (4/146; 2.74%), a non-RLN was discovered. All patients with a non-RLN were females aged 33–45 years.

A diagram showing the recorded voltage values of the subsequent steps of the performed thyroidectomy is displayed in Figure 6. In this figure, the signal from the VN before dissection and below the (ITA) was chosen as the 100% baseline value. Recorded voltages below or above it, taken at different points as mentioned above, are also plotted in the Figure.

The positive signal of ADNM-VN1 confirmed the presence of a right-sided non-RLN. A similar procedure was also conducted on the left side and no such signal could be obtained. This proves that the identification of non-RLN, conducted before continuing with the dissection, was successful. As the dissection went on, the nerves continued to generate EMG values, which are plotted in Figure 6. Non-RLN type 1 and type 2 were subsequently identified (Figure 3, Figure 4 and Figure 5) and their presence was confirmed by the positive EMG values. However, the amplitude of the EMG signal of the VN dropped below 50% of the baseline when it was recorded post-dissection. Since the data was recorded immediately after the dissection was over, the VN was almost numb and did not recover immediately. Therefore, the non-RLN could not generate any EMG data. After about 7 min and treatment with hydrocortisone, the VN started to recover, and an EMG signal could be recorded. This proves that the minimally invasive surgery was successful after using the ADNM protocol of stimulating precisely defined anatomical points; because of this, there was no nerve injury. The non-RLN was completely fine, and its function was intact.

This study is the first of its kind to utilize the NIM Vital^TM^ device to report the outcome in this large cohort, wherein the EMG signal of different anatomical points was measured. Neuromuscular blockade agents are typically used in anesthesia for endotracheal intubation. However, this may lead to the potentiation of these blockades, which leads to postoperative complications [12]. Therefore, in this study, no neuromuscular blocking agents were used; nonetheless, the thyroidectomy was successful. We were also able to identify the precise anatomical points where stimulation should be applied to identify the non-RLN without damaging its functionality, meaning that this is the first study in this field. As the prevalence of non-RLN is higher than previously expected, a thyroidectomy can prove to be difficult [3]. Our ADNM protocol for mapping the anatomical points can provide a guide for the surgeons so that a minimally invasive thyroidectomy can be conducted without the loss of function of non-RLN. However, our developed protocol can also be applied if the presence of non-RLN is not apparent. In other words, this protocol can be used with equal effectiveness in standard thyroidectomy procedures, where the surgeons will follow the anatomical points that have been precisely identified in the above protocol. Another important feature is that the thyroidectomy can be performed without using any neuromuscular blockade, to minimize the confounder with the EMG signal and its amplitude, and to save costs by avoiding muscle relaxants and antidotes and optimizing OT utilization, particularly in saving the time from the Kocher incision until the complete lobe resection, which varies, according to the surgeon, between 10 and 30 min, with an average of 18 min. The second point to minimize confounders or indirect damage to the VN or RLN is in postponing the use of energy or electrical sealing devices, as well as visualization and traction of the thyroid lobe, to reduce any possible risk of impaired EMG-amplitude caused by heat, electrical field, or mechanical traction. Those two steps will ensure the achievement of the real baseline EMG-amplitude. The use of an energy-based device can result in the paralysis of RLN, which can cause hoarseness after the operation [13]. The precise definition of those stimulation points is of great importance for both novices in thyroid surgery and new users of intraoperative neuromonitoring. The colored EMG value will allow the surgeon to constantly monitor his steps and correct his missteps in the case of any red signal points during the procedure. This correction can be made either by using less traction on the luxated lobe, by the revision of nerve entrapment in clips or sutures, preventing heat spread from vessel-sealing devices, or by tube correction, and changing the current settings on the NIM Vital^TM^ device. Furthermore, through the use of the ADNM protocol, a retrosternal goiter can be resected much more safely, using the ADNM mappings points before any forced mobilization of the retrosternal goiter.

This methodology includes several novel and important steps during thyroidectomy that will increase the chance of preventing voice dysfunction in the RLN, SLN, and non-RLN. For example, to protect the SLN, the thyroid gland was clamped very close to the upper pole vessels, being lifted from the prevertebral facia both distally and laterally. A vascular plane between the larynx muscle and upper pole vessels is of great importance when locating the SLN and protecting it from mechanical or thermal damage. Here, intermittent INOM testing for every portion that is planned for division is mandatory. The ligation should be performed after a clear capsule is closed, separating it into 2–3 lengths. The traction directions of the strap muscles and thyroid during surgery helped in the visualization of the nerves. Intraoperative thermal injury can occur due to the use of electricity or energy-based devices [14]. The energy-based devices were used with significant precautions. Damage to the SLN and non-RLN due to heat spread was prevented by cooling the area with cold saline after every heating device was used, in a radius of 2 cm to the nerve; otherwise, clips or 4.0 PDS stitches were used. The saline took away the excess heat and protected the nerves. Finally, by a sudden decrease in EMG-amplitude (point 7 in Figure 6) an intravenous injection of 200 mg hydrocortisone with 10–15 min of cooling of the field and the removal of all traction trocars on the situs can help to recover EMG-amplitude and continue the procedure in a relaxed OT-environment. Prior research has shown that individuals with temporary RLN-paresis recover faster corticosteroids are delivered intraoperatively, but the same research has failed to establish that a single dose of intraoperative corticosteroids can reduce the rate of temporary/permanent RLN- paresis after surgery [15,16].

Our study is limited by its retrospective nature and its relatively small patient sample. Controlled, randomized studies with larger patient samples are needed to compare the ADNM technique to the gold standard of thyroidectomy. The low and uncertain prevalence of non-RLN is another obstacle.

## 4. Conclusions

To our knowledge, this is the first study describing the clinical benefit of using the new NIM Vital™ (Medtronic Xomed, Inc., Jacksonville, FL, USA) in a large patient cohort, without the use of any neuromuscular blockade for intubation.

The developed ADNM protocol was successful in stimulating precisely defined anatomical points on the VN and SLN to identify the non-RLN accurately during a minimally invasive thyroidectomy. These stimulation points helped guide the surgeon for the early detection of non-RLN. Obtained EMG signal values were compared with the baseline value of VN pre-dissection, which was taken as 100%. The post-dissection values of SLN and VN fell well below the baseline, to the point where the nerves were considered numb. However, after treatment with hydrocortisone, the VN recovered, and voice functionality was intact. During the surgery, besides the specified anatomical points, novel steps were taken that increased the chance of the visualization of nerves and preservation of their functions, without using any endoscopic optical instruments, such as the safe ligation of upper pole vessels, cooling of the same area to prevent thermal injury, specified traction direction, and hydrocortisone application for quick recovery of the nerves. Our developed ADNM protocol has the potential to improve cost-effectiveness by better OT utilization through fast and secure surgery, reducing expensive muscle relaxants and antidotes. Even without the presence of non-RLN, the same principles should be applied. Endocrine surgeons can monitor their surgical procedures in real time by observing the EMG values, which will act as a guide from beginning to end.

## Figures and Tables

**Figure 1 jcm-11-05677-f001:**
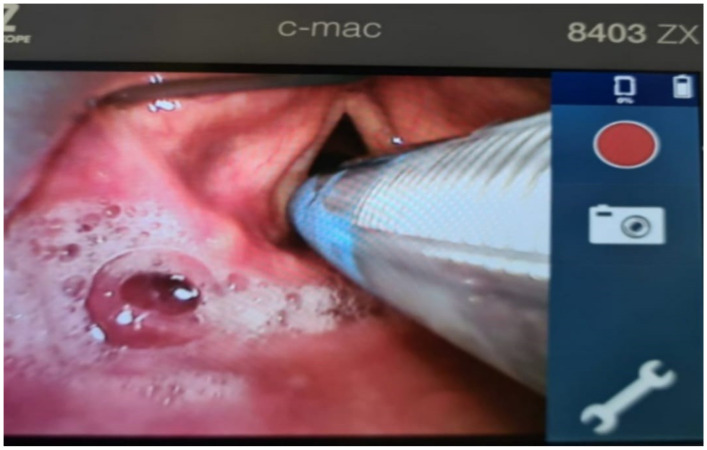
Videolaryngoscopic placement of the EMG-endotracheal tube using C-Mac (Stortz), without a neuromuscular blockade. The blue electrode touches the mucosa of both vocal cords to ensure a proper EMG signal during stimulation.

**Figure 2 jcm-11-05677-f002:**
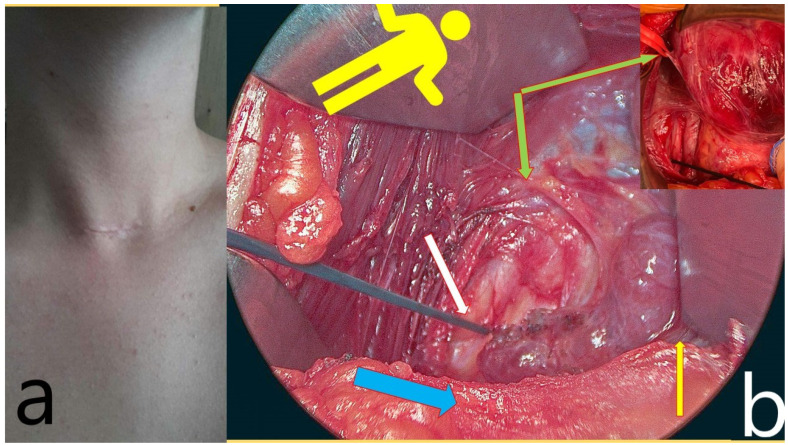
(**a**) A small 3-cm incision is used for minimally invasive open thyroidectomy. (**b**) The yellow figure indicates the patient’s orientation on the Operation table. The white arrow shows the vagal nerve stimulation point ADNM-VN1, the yellow one shows the strap muscle, separated from the sternocleidomastoid but intact over the anterior thyroid capsule, which is retracted medially, and the green arrow shows the middle thyroid vein entering the internal jugular vein (upper right corner, in the red vessel loop. Finally, the blue arrow shows the manubrium of the sternum.

**Figure 3 jcm-11-05677-f003:**
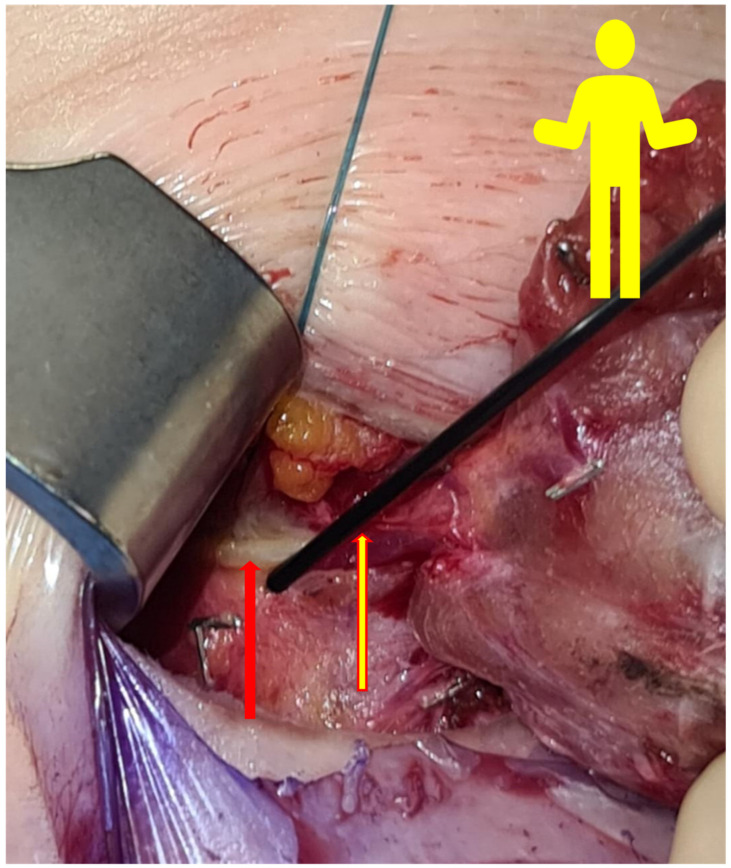
The yellow figure simulates the patient’s position on the OT table. The red arrow shows the first non-RLN type 1. The yellow arrow shows the stimulation probe of the NIM Vital™ (Medtronic Xomed, Inc., Jacksonville, FL, USA).

**Figure 4 jcm-11-05677-f004:**
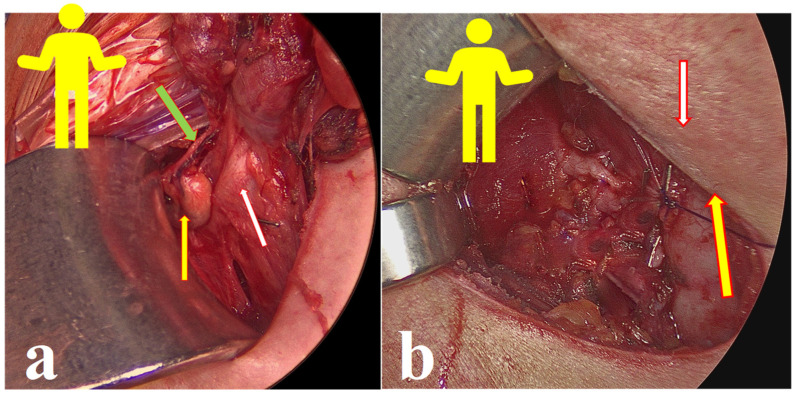
(**a**) The yellow figure simulates the patient position on the OT table. The white arrow shows the second non-RLN type 1. The yellow arrow shows the upper parathyroid and the green arrow indicates the blood supply of the upper parathyroid. (**b**) The yellow arrow shows non-RLN type 1 and the white arrow shows the upper parathyroid.

**Figure 5 jcm-11-05677-f005:**
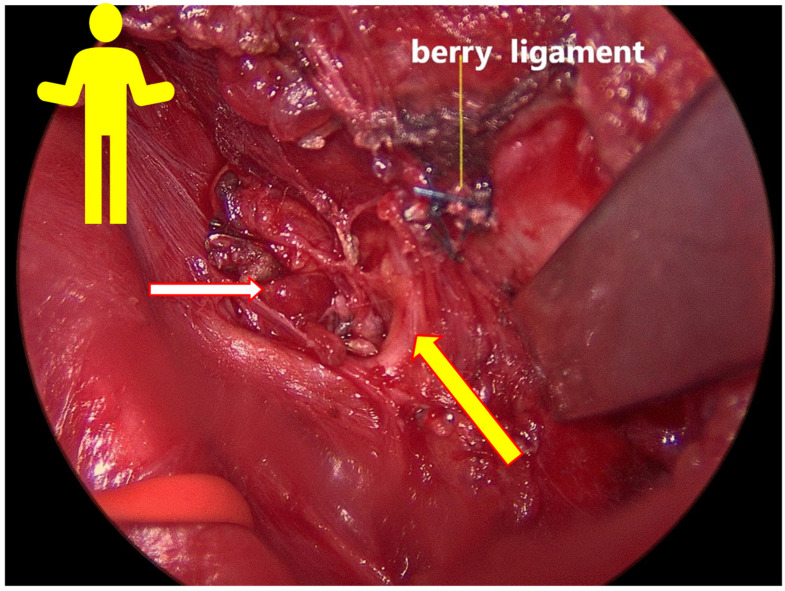
The yellow figure simulates the patient’s position on the OT table. The yellow arrow shows non-RLN type 2, and the white arrow shows the upper parathyroid.

**Figure 6 jcm-11-05677-f006:**
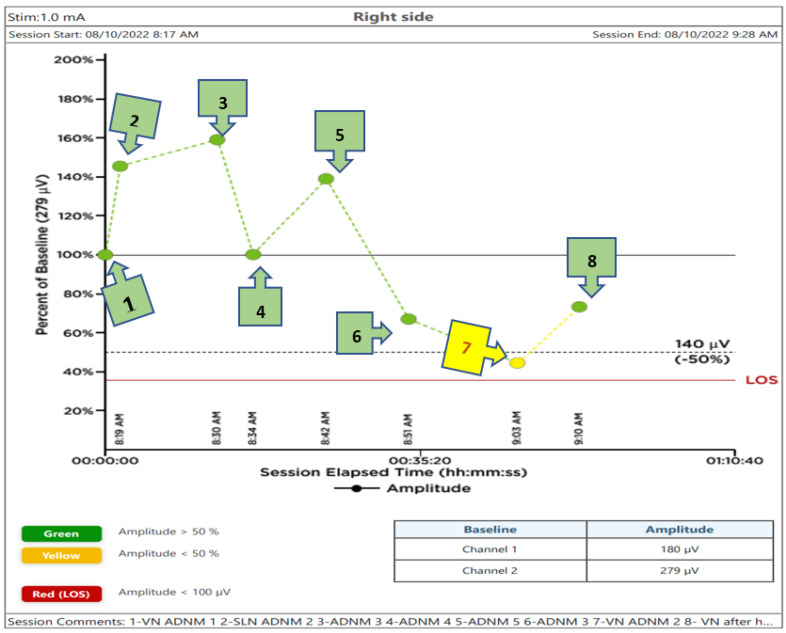
The steps of the Abu Dhabi Neural Mapping (ADNM) diagram during a right-sided hemithyroidectomy, using the intraoperative neuromonitoring system, NIM Vital™, with a stimulation current of 1 mA. Here, the number indicates the following signals: 1. Vagal nerve pre-dissection below the crossing point of RLN, with ITA as the baseline EMG-amplitude and without any manipulation of the thyroid (ADNM−VN1). 2. EMG-amplitude of an SLN pre-dissection at the level of upper pole vessel division (ADNM−SLN1). 3. EMG-amplitude of an RLN pre-dissection below-crossing with ITA (ADNM-RLN1). 4. EMG-amplitude of an RLN pre-dissection above-crossing with ITA (ADNM−RLN2). 5. EMG-amplitude of RLN pre-dissection at the berry ligament (ADNM−RLN3). 6. EMG-amplitude RLN post-dissection below-crossing with ITA (ADNM−RLN4). 7. EMG-amplitude of VN post-dissection at the level of the larynx incisura, which is below 50% of the baseline (ADNM −VN2). 8. EMG-amplitude after 200 mg hydrocortisone i.v. and, after cooling for 5 min, recovery of EMG-amplitude (ADNM−VN2 recovery). Green points represent EMG amplitude above 50% of the baseline pre-dissection; yellow points were used for EMG amplitude between 50 and 20% of the baseline pre-dissection; red points indicate an EMG-amplitude below 20% of the baseline pre-dissection.

## Data Availability

The data that support the findings of this study are available from the corresponding author upon reasonable request.

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
