# Peer review of "Abu Dhabi Neural Mapping (ADNM) during Minimally Invasive Thyroidectomy Enables the Early Identification of Non-Recurrent Laryngeal Nerve and Prevents Voice Dysfunction"

_jcm, 2022, doi:10.3390/jcm11195677_

Round 1
Reviewer 1 Report
This is an excellent study. It is of interest because it describes a specific protocol for nerve monitor use that can be repeated by other surgeons. Evaluation of the efficacy of nerve monitoring in thyroid surgery has been limited by highly variable protocols of use by different surgeons. My only concern is in the description of the infusion of intraoperative steroids that were used when the response to vagal stimulation was below 50% after dissection. The use of hydrocortisone infusion was not mentioned in the initial description of the protocol and I ask the authors to be more specific about indications for use. Should steroid be infused only if the the stimulation response drops below a certain threshold, in every case, or in every case where a non-recurrent nerve is suspected and found? Use of steroids has not been "routine" relative to monitoring issues in thyroid surgery and it's not clear whether the authors claim to have "proven" that steroids help the recovery of stimulation response in all cases. It's a reasonable suggestion but there needs to be more clarity about how the steroid infusion fits in to the ADNM protocol.
Author Response
Dear Reviewer,
Thank You Very Much Please accept this amended version of our manuscript "Abu Dhabi Neural Mapping (ADNM) during Minimally Invasive Thyroidectomy Predicts early Identification of Non-Recurrent Laryngeal Nerve & Prevents Voice Dysfunction" for publication in the Journal of Clinical Medicine. Thank you for taking the time to review our work and for the helpful suggestions and edits that you and the other reviewers provided. The vast majority of the reviewers' recommendations have been implemented. Those alterations have been marked up in the text. A detailed answer to the reviewers' remarks and questions is provided in blue below. Page references are to the edited, tracked-changes version of the manuscript.
Recommendations from Reviewers to the Authors:
Should steroid be infused only if the the stimulation response drops below a certain threshold, in every case, or in every case where a non-recurrent nerve is suspected and found?
We agree with the reviewer that routine steroid administration is controversial, and that prior research has failed to show that intraoperative corticosteroids injection can reduce the incidence of temporary/permanent RLN- paresis following surgery. However, corticosteroids have been shown to speed up recovery in patients with transient RLN- paresis (ling Wang et al., World J Surg., 2006 Mar;30(3):299-303). At the location of injury, further along the length of the injured axons, and at the peripheral terminals themselves, there may be an influx of inflammatory and immune cells after damage to a peripheral neuron (Archives of Oral Biology, https://doi.org/10.1016/j.archoralbio.2006.08.015). When we detect a decrease in signal, regardless of whether or not a non-recurrent nerve is present, we immediately administer a steroid to protect the nerve fibers. Applying cold saline to the operation area relieves mechanical pressure on the gland for 15 minutes (during which the signal weakens). Micro injury caused by traction or heat distribution around the RLN can be mitigated by injecting a single dose of steroid into the region around the nerve. A single dose of intravenous 4mg Dexamethasone provided intraoperatively by loss of Signal after Thyroidectomy was sufficient to restore EMG signal, according to a recent study (Donatini G, etal World J Surg. 2020 Feb;44(2):417-425. doi: 10.1007/s00268-019-05295-2. PMID: 31741073). These additions can be found in the revised manuscript's expanded discussion.
Best Regards
I.Hassan

Reviewer 2 Report
The authors evaluated the accuracy of a neuromonitoring protocol for early identification of non recurrent laryngeal nerve.
In case of negative signal did you try to stimulate the nerve proximately?
Limitations of the study have to be discussed better.
References can be improved.
Author Response
Dear Reviewer,
Thank You Very Much Please accept this amended version of our manuscript "Abu Dhabi Neural Mapping (ADNM) during Minimally Invasive Thyroidectomy Predicts early Identification of Non-Recurrent Laryngeal Nerve & Prevents Voice Dysfunction" for publication in the Journal of Clinical Medicine. Thank you for taking the time to review our work and for the helpful suggestions and edits that you and the other reviewers provided. The vast majority of the reviewers' recommendations have been implemented. Those alterations have been marked up in the text. A detailed answer to the reviewers' remarks and questions is provided in blue below. Page references are to the edited, tracked-changes version of the manuscript.
Recommendations from Reviewers to the Authors:
Have you attempted close nerve stimulation in the event of a negative signal?
I appreciate you bringing that up. Although we agree that testing of the VN more proximal may prove useful, we are wary of doing so for the reasons listed below. It will first necessitate more proximal dissection in minimally invasive surgery in order to access the vagal nerve at our defined proximal point, which is the level incisura of the larynx. This is a challenging step because the entire lobe remains in situ during the minimally invasive procedure's with a narrow working space, which increases the risk of bleeding and neural injury and provides no additional benefit, since we will continue as for existing non-RLN by negative signal on our defined VN1 and intact technical and anaesthetic circumstances after confirming the device's functionality on the left side.
Second, we determined to stick to our new operating procedures of the aforementioned protocol at all times in order to ensure that our Protocol was successful in its secondary goal of teaching our Trainee. That means if they begin on the left side and get no signal on our VN1, the doctors will have to rule out technical or anaesthetic errors by evaluating the right side.
(x) The English grammar and syntax are acceptable; a quick spell check would be helpful.
Two typos on page 4 have been fixed: the word 'ligasure' has been changed to 'LigaSureTM,' and the word 'dame' has been changed to 'damage,' on line 24.
The study's limitations need to be examined more thoroughly.
The retrospective design of our study and the modest size of our patient sample are its main flaws. More robust controlled, randomized studies comparing the ADNM-technique to the gold standard of thyroidectomy are required. Another challenge is that the prevalence of Non RLN is low and unknown.
The citations could be better.
The new references 15 and 16 have been added.
Best Regards
I.Hassan
